# Is There a Correlation between Endoscopic Sinus Surgery and Improvement in Erectile Dysfunction?

**DOI:** 10.3390/jcm12206626

**Published:** 2023-10-19

**Authors:** Antonio Romano, Umberto Committeri, Vincenzo Abbate, Marco Sarcinella, Francesco Maffia, Simona Barone, Stefania Troise, Giovanni Salzano, Riccardo Nocini, Paola Bonavolontà, Giovanni Dell’Aversana Orabona

**Affiliations:** 1Maxillofacial Surgery Unit, Department of Neurosciences, Reproductive and Odontostomatological Sciences, “Federico II” University of Naples, Via Pansini 5, 80100 Naples, Italy; romano.antonio1972@gmail.com (A.R.); umbertocommitteri@gmail.com (U.C.); vincenzo.abbate@unina.it (V.A.); m.sarcinella@studenti.unina.it (M.S.); simo_baro@yahoo.it (S.B.); stefy.troise@gmail.com (S.T.); giovannisalzanomd@gmail.com (G.S.); paola.bonavolonta@unina.it (P.B.); giovanni.dellaversanaorabona@unina.it (G.D.O.); 2Unit of Otolaryngology, Head and Neck Department, University of Verona, 37134 Verona, Italy; riccardo.nocini@gmail.com

**Keywords:** chronic rhinosinusitis, endoscopic sinus surgery, nasal obstruction, erectile dysfunction

## Abstract

Background: In recent years, the focus on respiratory disorders has increased, notably on chronic rhinosinusitis (CRS), an inflammatory condition of the upper airway that can significantly impact one’s quality of life. Interestingly, CRS has emerged as a potential comorbidity in erectile dysfunction (ED). This study aims to assess the impact of endoscopic sinus surgery for CRS on sexual function. Materials and Methods: The authors conducted a prospective study of patients who visited their clinics for chronic rhinosinusitis between June 2018 and June 2022. The study involved 53 patients aged between 40 and 70 years who were treated for CRS with nasal polyps (CRSwNP) and CRS without nasal polyps (CRSsNP). Preoperative and postoperative assessments were performed using the Nasal Obstruction Symptoms Evaluation Score (NOSE score) and the 5th International Index of Erectile Function Score (IIEF-5 score) to evaluate potential improvements in sexual function following endoscopic sinus surgery. Results: Before surgery, the average NOSE score was 72.6, which decreased to 24.9 postoperatively. The average preoperative IIEF-5 score was 16.35, while the postoperative average increased to 19.52. Statistical analysis revealed a significant improvement in erectile function for penetration (*p*-value = 0.024) and overall satisfaction after intercourse (*p*-value < 0.001) regarding the degree of nasal obstruction. Conclusion: This study underscores the potential benefits of treating chronic obstructive upper airway diseases such as sinusitis in improving the sexual outcomes of patients clinically diagnosed with erectile dysfunction.

## 1. Introduction

Nasal airway disorders are a diverse group of conditions influenced by factors such as geography, demographics, and individual health. These disorders encompass structural issues like deviated septum and turbinate hypertrophy, as well as allergic rhinitis, chronic sinusitis without nasal polyps (CRSsNP), and chronic sinusitis with nasal polyps (CRSwNP), among others [1]. CRS is a prevalent inflammatory condition affecting the nasal and sinus passages, characterized by persistent inflammation of the paranasal sinuses lasting at least 12 weeks. Its prevalence ranges from 5% to 12% in the general population, with symptoms including nasal congestion, facial pain, headaches, reduced sense of smell, and nasal discharge, causing chronic suffering. CRS negatively impacts sleep quality, work productivity, and overall health-related quality of life, resulting in a substantial economic burden [2]. Recently, more attention has been paid to the link between nasal airway obstruction and concurrent erectile dysfunction (ED), defined as the inability to achieve a satisfactory erection for sexual intercourse [3,4,5,6]. Some studies, such as the European Male Ageing Study (EMAS), have reported ED prevalence ranging from 6% to 64% among men aged 40–79 years, increasing with age, with an average prevalence of 30% [7]. Some reports suggest a potential connection between CRS, nasal polyposis (NP), and ED [8,9,10]. Patients with CRS have a higher risk of stroke and acute myocardial infarction (AMI), indicating how CRS may affect blood oxygenation [11,12]. Moreover, CRS and NP can lead to symptoms like hyposmia and compromise the physiological olfactory function, a vital aspect of the reproductive sphere. A few recent studies have indicated positive effects on erectile dysfunction from the restoration of normal airflow following surgery [13,14,15]. This study’s objective was to investigate the impact of functional endoscopic sinus surgery (FESS) on the sexual function of individuals with ED who also have CRS, with or without polyps.

## 2. Materials and Methods

### 2.1. Preoperative Assessment

This is a prospective study of a cohort of patients admitted to the Department of Maxillo-Facial Surgery at Federico II University Hospital for nasal airway obstructions between June 2018 and June 2022. The ethical approval was obtained from the Institutional Review Board for Clinical Research at the Unit of Maxillofacial Surgery of Napoli Federico II, protocol number (88/20). The study was conducted following the Declaration of Helsinki. A total of 280 patients were selected during their first visits to our clinics and evaluated according to the following inclusion criteria:Male patients aged between 40 and 70 years old;Nasal obstruction caused by CRS and or NP confirmed by nasal endoscopic examination and paranasal sinus CT scan;Patient with indication for FESS;Personal perception of the nasal obstruction assessed through the Nasal Obstruction Symptom Evaluation (NOSE) scale questionnaire before surgery and 3 months after (development and validation of the Nasal Obstruction Symptom Evaluation (NOSE) scale);Clinical diagnosis of erectile dysfunction by using the 5th International Index Erectile Function Score (the use of the simplified International Index of Erectile Function (IIEF-5) as a diagnostic tool to study the prevalence of erectile dysfunction) [16].

To avoid biases between cardiocirculatory pathologies and erectile dysfunction, patients affected by risk factors such as obesity (BMI > 24.9 kg/m^2^), hypertension, diabetes mellitus, penis deformities, peripheral neuropathies, patients undergoing testosterone therapies or therapies for andrological diseases, or remote anamnesis for prostatic or nasal or sinus surgery were excluded from the study. All systemic and peripheral risk factors that could worsen or cause a mild state of erectile dysfunction were eliminated.

The International Index Erectile Function Score (IIEF-5) (Figure 1) and Nasal Obstruction Symptom Evaluation (NOSE) (Figure 2) were submitted routinely before surgery and at a 3-month follow-up after surgery.

Out of a total of 280 excluded patients, 226 were excluded because they did not have an ED diagnosis, 20 did not undergo FESS, and 34 did not complete the IIEF-5 questionnaire postoperatively. A total of 53 patients met the inclusion criteria and were included in the study. The control group was composed of healthy people, not affected by nasal obstruction and erectile dysfunction, adopting the exclusion criteria used for the treated patients. The control group was used to compare the study group to a hypothetical selected healthy population. This strategy was used to assess the discrepancy between the normal scores of the enrolled patients in the preoperative phase and to evaluate the outcome obtained in the postoperative phase. A total of 49 patients were chosen as control cases and ranged in age from 40 to 79 years old. All patients were subjected to a comprehensive clinical evaluation including a nasal endoscopy examination and underwent a computerized tomography (CT) scan of the paranasal sinus. The diagnosis of chronic upper airway obstruction was confirmed by the nasal endoscopy and the paranasal CT (110kV, 210 mAs, 3 mm slice thickness). All patients underwent functional endoscopic sinus surgery (FESS) performed by the same surgical team, and the operations were all carried out with general anesthesia.

### 2.2. Surgical Procedure

The surgical treatment of chronic sinusitis involved the clearing of the osteomeatal complex (OMC) before uncinectomy and bullectomy procedures. Anterior and anteroposterior ethmoidectomies were performed, when necessary, with or without sphenoidotomy. When the pathology required the removal of inflammatory tissue from the frontal sinus, senotomy of the frontal sinus was carried out with the removal of the inflamed and/or polypoid tissue along with balloon-assisted senoplasty. The surgical instruments adopted for the operations included microdebriders (Stryker, Core ESSx, UK Ltd., Berkshire, UK) and a balloon (Stryker, Xpr ESS Ultra ENT dilatation system, UK Ltd., Berkshire, UK). When needed, a septoplasty was carried out utilizing the Cottle technique. In the case of turbinate hypertrophy, a turbinotomy was executed.

### 2.3. Statistical Analysis

Data were described as mean and standard deviation. Differences between means for continuous variables were analyzed with the Wilcoxon signed-rank sum test for paired samples and with the Kruskal–Wallis test for independent samples, as appropriate. Mixed-effect multiple linear regression using time as a random effect was used to investigate the effect of the factors on the outcomes of interest. For all analyses, a *p* < 0.05 was considered significant. Analyses were performed using the statistical software R, version 4.0.3.

## 3. Results

The mean age was 51.3 ± 3.5 years old and the average BMI was 22.9 ± 1.3 kg/m^2^. A total of thirteen patients suffered from chronic pansinusitis, twenty from unilateral maxillary and ethmoidal chronic sinusitis, nine suffered from bilateral chronic sinusitis and twelve suffered from unilateral maxillary chronic sinusitis. Among 53 patients, 18 had concha bullosa, 39 were subjected to an inferior turbinectomy, and 20 to an associated septoplasty. The preoperative average NOSE score was 72.6. The average postoperative value decreased significantly to 24.9.

The pre and postoperative results related to the IIEF-5 questionnaire are expressed in Table 1. The average preoperative IIEF-5 value was 16.35, whereas the average postoperative value increased to 19.52. The pre and postoperative results related to the NOSE score questionnaire are expressed in Table 2.

All evaluated variables are displayed in Table 3, Table 4, Table 5, Table 6 and Table 7. In addition, through multiple linear regression, it can be shown that the different degree of nasal obstruction is statistically significant toward improving “Erection hard enough for penetration” (*p*-value = 0.024) and the “Grade of satisfaction” (*p*-value < 0.001). In addition, two other variables were statistically significant for improved initial penetration: improved breathing during sleep (*p*-value = 0.008) and during physical activity (*p*-value = 0.006).

## 4. Discussion

The primary aim of this study was to investigate the potential correlations between chronic inflammatory upper airway diseases and erectile dysfunction (ED). Patients included in the study underwent functional endoscopic sinus surgery (FESS) to eliminate inflammatory tissue within the sinuses and restore functional nasal anatomy for proper breathing. Postoperatively, all patients experienced notable improvements in nasal breathing, as evidenced by a reduction in the average Nasal Obstruction Symptoms Evaluation Score (NOSE score) from 72.6 to 24.9. Importantly, the study also observed an enhancement in the patients’ sexual sphere following FESS, with the International Index of Erectile Function Score (IIEF-5 score) increasing from an average of 16.35 to 19.52. Using multiple linear regression analysis, statistically significant relationships were identified between the improvement in the variable “Erection hard enough for penetration” and factors such as “Nasal obstruction” (*p*-value = 0.024), “Trouble sleeping” (*p*-value = 0.008), and “Unable to get air” (*p*-value = 0.006). Additionally, a significant association between the improvement in the variable “Nasal obstruction” and the variable “Grade of satisfaction” (*p*-value < 0.001) was found.

It is essential to consider the systemic consequences that can result from the obstruction of the paranasal sinuses. The existing literature has already described how chronic obstructive diseases can disrupt normal blood oxygenation and consequently impact pulmonary blood pressure [14,17,18]. Specifically, Fidan et al. demonstrated that even minor nasal obstruction can affect physiological pulmonary ventilation [14]. When breathing through the nose during expiration, the airflow slows down due to nasal anatomy, leading to improved oxygen release through red blood cells. Conversely, chronic upper airway obstruction often prompts mouth breathing, especially during sleep, causing increased respiratory frequency and decreased oxygen absorption. This compensatory response includes tachypnea, elevated pulmonary blood pressure, and peripheral vascular constriction, ultimately culminating in systemic consequences that may also affect penile vascularization and the sexual sphere [7,18,19,20]. Furthermore, nasal airway obstruction, when associated with snoring and sleep apnea (interruptions in regular breathing), can constitute obstructive sleep apnea syndrome (OSAS). Nocturnal hypoxemia, a significant risk factor for arterial hypertension and cardiovascular complications, underscores the importance of normal respiratory function during sleep [19]. Studies, such as those by Fidan and Aksaka, have highlighted the potential for improvements in mean pulmonary arterial pressure (mPAP) after surgery to correct septum deviation. This suggests a link between nasal obstruction and pulmonary vascular dynamics [14]. Another complication associated with chronic rhinosinusitis is olfactory dysfunction resulting from inflammation and nasal congestion that affect the olfactory epithelium responsible for detecting odors. This can have implications for human ingestion, including the modulation of appetite, and even aspects of mammalian sexual behavior. Olfactory regulation plays a critical role in social communication, reproductive behavior, and mate selection [21,22,23].

Erectile dysfunction etiology has been historically identified as organic, psychogenic, and mixed; traditionally considered predominantly psychogenic, it is increasingly recognized as having organic origins in more than 80% of cases [24]. Vasculogenic factors, which encompass issues related to the blood supply, particularly arterial inflow disorders, and venous outflow abnormalities, are the most common causes of organic ED. Reduced blood flow and arterial insufficiency play pivotal roles in this multifactorial etiology [7,20,21,22].

However, it’s important to note that ED of vascular origin does not typically arise solely from high blood pressure; it often follows alterations in the vascular wall, particularly a reduction in elasticity as a response to elevated pressure. Arterial hypoxia in the corpus cavernosum results in decreased levels of prostaglandin E1, which ordinarily inhibits the action of transforming growth factor beta 1 (TGFB1). This hypoxia encourages collagen deposition, leading to reduced vessel elasticity. When the ratio of smooth muscle to collagen decreases and collagen content increases, the ability of the corpora cavernosa to constrict the subtunical veins diminishes, ultimately causing veno-occlusive dysfunction [7]. Beyond these vascular factors, there are other aspects to consider for a comprehensive understanding of ED. Hypoxia induces vasoconstriction and significantly reduces the activity of nitric oxide synthetase, which is a risk factor that limits nitric oxide (NO) production in the corpora cavernosa. Lindberg et al. [25] reported that patients with CRS had lower concentrations of nasal nitric oxide (NO) compared to their healthy counterparts. Ragab et al. [26] observed that levels of nasal NO increased following medical and surgical treatment for CRS. These findings suggest a link between NO levels and CRS treatment. Additionally, various studies in the literature have explored other factors that may contribute to the pathophysiology of both ED and CRS. For instance, the role of C-reactive protein (CRP) has been examined. Patients diagnosed with ED who also underwent Doppler examination showed significant increases in serum CRP levels [27]. Giugliano et al. [28] found that patients with ED and a high body mass index (BMI) had higher CRP values compared to patients with a similar BMI but without an ED diagnosis (*p* < 0.05). Similarly, an increase in CRP levels was observed in patients diagnosed with CRS compared to control cases [29,30].

In the case of obesity, there is an infiltration of immune cells into the adipose tissues, with resultant leptin and insulin resistance. All of that induces lipotoxicity and oxidative stress in peripheral tissues as well as generates an inflammatory response in the hypothalamus like the decreased secretion of gonadotropin-releasing hormone (GnRH), which reduces the levels of testosterone. Another mechanism reducing testosterone levels seen in obesity is elevated aromatase by hypertrophied adipose tissue; this increases the conversion of testosterone to estradiol, and elevated estradiol suppresses GnRH in the hypothalamus via a negative feedback mechanism [31]. Hypertension and erectile dysfunction are closely intertwined diseases, which have endothelial dysfunction as a common base. During hypertension and/or erectile dysfunction, the disturbance of endothelium-derived factors can lead to an increase in vascular smooth muscle (VSM) contraction. Hypertension can lead to erectile dysfunction as a consequence of high blood pressure (BP) or due to antihypertensive treatment [32]. Poor glycemic control, testosterone deficiency, and peripheral arterial disease are the risk factors for ED in diabetic subjects [33,34].

In the past, Hirshoren et al. [35] reported higher CRP levels in patients with severe sinusitis, suggesting that local mucosal inflammation in CRS may contribute to elevated inflammation factors, potentially playing a role in the pathogenesis of ED. Moreover, tumor necrosis factor-alpha (TNF-α) appears to have a pathophysiological role in ED. Two studies have indicated that patients diagnosed with CRS-NP exhibit higher levels of TNF-α in both nasal cavity mucus and blood samples [36,37].

In summary, while vascular factors are a common cause of ED, it is essential to consider other aspects, such as the role of NO, CRP, and TNF-α, as well as their potential links to conditions like CRS. A holistic understanding of these factors can provide valuable insights into the pathophysiology of ED and related conditions [25,27,29,30,35,36,37]. Beyond the well-established psychogenic factors, ED can also have organic etiologies, with vascular issues being the most common. The present study sheds light on the potential interplay between chronic upper airway diseases and ED, emphasizing the importance of addressing nasal obstruction as a potential contributor to ED pathogenesis. Factors such as altered blood oxygenation, systemic inflammation, and olfactory dysfunction may all play roles in this complex relationship, warranting further investigation.

The limitations of this study are represented by the small sample size, the complexity in the accurate assessment of outcomes related to improved blood oxygenation, the multifactorial nature of the pathologies examined, and the weight of the psychological component related to erectile dysfunction.

## 5. Conclusions

This study provides evidence of how the treatment of chronic obstructive upper airway diseases can improve the sexual outcomes of patients clinically diagnosed with erectile dysfunction. Notably, the average postoperative NOSE score significantly decreased to 24.9, indicating substantial improvements in nasal obstruction. Furthermore, the average preoperative IIEF-5 score was 16.35, while the average postoperative value increased to 19.52. Consistent with the previous literature, there was a significant increase in all domains of erectile dysfunction assessed by the questionnaire following surgical treatment.

This research highlights the importance of considering nasal obstruction and its systemic effects as potential contributors to erectile dysfunction. Erectile dysfunction etiology has been identified as organic, psychogenic, and mixed. Our findings suggest that addressing nasal obstruction through surgical intervention can yield significant improvements in both nasal function and sexual outcomes because of the many consequences on the respiratory, vascular, and therefore systemic spheres that chronic upper airway obstruction can cause. This complex relationship warrants further investigation, particularly regarding factors like altered blood oxygenation, systemic inflammation, and olfactory dysfunction.

## Figures and Tables

**Figure 1 jcm-12-06626-f001:**
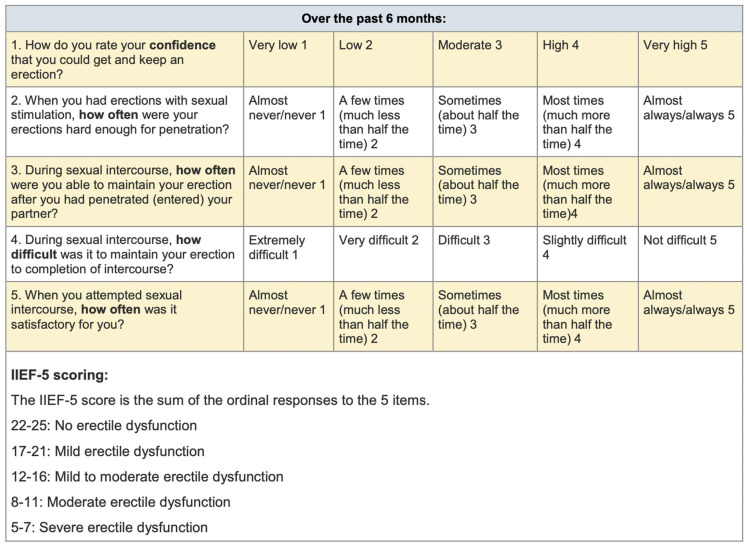
Simplified International Index of Erectile Function (IIEF-5) IIEF-5 SCORE. Domains analyzed are: achieving and keeping an erection, an erection hard enough to penetrate, keeping an erection after penetration, maintaining an erection, and satisfactory sex. Answers are classified from 1 “rarely or never” to 5 “almost always, always”.

**Figure 2 jcm-12-06626-f002:**
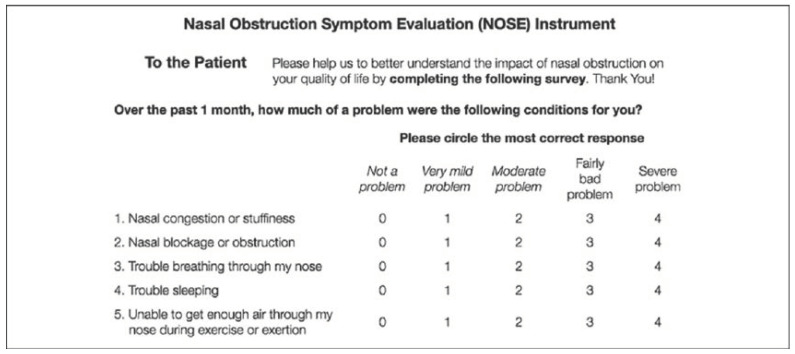
Nasal Obstruction Symptoms Evaluation (NOSE) instrument. Domains analyzed are nasal congestion or stuffiness, nasal blockage, trouble nose breathing, trouble sleeping, and difficulty in nose breathing during exercise. Answers are classified from 0 “not a problem” to 4 “severe problem”.

**Table 1 jcm-12-06626-t001:** Highlights of the average of the results for each question proposed by the IIEF-5 and NOSE score questionnaires. The degree of significance of the preoperative vs. postoperative sample and preoperative sample vs. control cases is then demonstrated. All results are statistically significant (*p* value < 0.001 except for “Grade of satisfaction” investigated by IIEF-5). In this case, it is shown that patients after undergoing endoscopic surgery demonstrated marked improvements in both ventilation and quality of erectile function.

Age	Question n° 1		Question n° 2		Question n° 3		Question n° 2		Question n° 3		IIEF-5 PRE	IIEF-5 POST
	PRE	POST	PRE	POST	PRE	POST	PRE	POST	PRE	POST		
**40**	4	5	2	3	2	3	3	4	4	4	15	19
**40**	4	4	3	5	5	5	4	5	4	4	20	23
**40**	3	3	2	2	3	3	3	3	4	5	15	16
**40**	4	5	3	5	3	5	4	5	4	4	18	24
**40**	3	4	2	2	3	4	3	4	5	5	16	19
**40**	3	4	3	5	4	4	4	5	5	5	19	23
**41**	4	5	3	5	3	3	4	5	4	5	18	23
**41**	5	5	2	2	4	4	5	5	4	4	20	20
**42**	5	5	2	3	3	5	5	5	4	4	19	22
**42**	5	5	3	5	2	4	4	5	5	5	19	24
**42**	3	5	5	5	2	3	3	5	5	5	18	23
**42**	3	4	1	2	3	3	2	2	5	5	14	16
**43**	4	5	5	5	3	3	5	5	4	4	21	22
**44**	3	5	5	5	3	5	3	3	4	4	18	22
**44**	4	5	3	5	3	3	3	5	4	4	17	22
**44**	2	5	5	5	2	2	5	5	5	5	19	22
**46**	3	5	1	2	2	3	2	2	5	5	13	17
**46**	3	5	5	5	3	4	5	5	5	5	21	24
**46**	2	5	5	5	3	5	3	3	4	4	17	22
**47**	3	5	3	5	2	2	3	5	4	4	15	21
**48**	4	5	5	5	3	3	5	5	4	4	21	22
**49**	5	5	1	2	2	2	2	2	3	5	13	16
**49**	3	5	5	5	2	3	5	5	4	4	19	22
**49**	2	4	5	5	3	4	3	3	5	5	18	21
**50**	1	4	5	5	3	5	5	5	4	5	18	24
**50**	2	4	5	5	3	3	3	3	3	5	16	20
**50**	3	4	3	5	4	4	5	5	4	4	19	22
**50**	3	5	5	5	5	5	3	5	5	5	21	25
**50**	3	4	1	2	4	4	2	2	4	4	14	16
**51**	4	5	1	2	5	5	3	3	3	3	16	18
**51**	3	5	5	4	3	5	4	4	4	4	19	22
**52**	4	4	4	5	4	4	3	3	5	5	20	21
**53**	5	5	1	2	3	3	3	3	4	4	16	17
**53**	3	5	5	5	3	3	3	4	3	4	17	21
**53**	3	4	4	5	2	3	3	3	4	5	16	20
**54**	1	2	1	3	2	3	3	3	5	4	12	15
**55**	2	3	2	3	3	3	4	4	4	4	15	17
**57**	3	3	4	5	2	3	3	3	5	5	17	19
**59**	1	3	2	3	2	3	4	4	4	4	13	17
**60**	2	2	4	5	3	3	3	5	3	3	15	18
**61**	2	3	3	3	4	4	4	4	4	4	17	18
**61**	2	3	1	2	3	3	3	3	4	4	13	15
**61**	1	4	2	3	3	5	5	5	3	3	14	20
**62**	2	3	3	3	2	4	3	4	5	5	15	19
**62**	3	4	1	2	2	4	3	3	3	3	12	16
**63**	2	5	1	2	2	3	3	3	4	4	12	17
**63**	3	3	2	3	4	4	2	3	5	5	16	18
**63**	3	3	3	3	2	4	3	4	4	4	15	18
**64**	4	4	1	2	3	3	3	3	3	3	14	15
**65**	3	4	1	3	2	2	3	3	4	4	13	16
**66**	2	2	2	3	3	2	3	4	5	5	15	16
**66**	1	1	3	3	2	3	3	4	4	5	13	16
**70**	3	3	1	2	1	2	3	3	3	4	11	14
**51.320**	2.9811	4.0943	2.9245	3.6981	2.8679	3.5283	3.4528	3.8867	4.1320	4.3207	16.3584	19.5283

**Table 2 jcm-12-06626-t002:** Average results obtained from questionnaires in patients undergoing surgery and control cases. Data is described as means ± SD. Significant values are marked in bold. * Computed with Wilcoxon signed-rank test. ** Computed with Mann–Whitney U test.

Variable	Pre, N = 53	Post, N = 53	Controls, N = 49	*p*-Value * (Cases Pre vs. Post)	*p*-Value ** (Cases Pre vs. Controls)
**Nasal stuffiness**	3.02 (0.93)	1.34 (1.06)	0.39 (0.49)	**<0.001**	**<0.001**
**Nasal obstruction**	3.19 (0.74)	0.51 (0.64)	0.35 (0.48)	**<0.001**	**<0.001**
**Trouble breathing through the nose**	2.94 (0.82)	0.77 (0.67)	0.12 (0.33)	**<0.001**	**<0.001**
**Trouble sleeping**	2.25 (1.22)	1.00 (0.88)	0.29 (0.46)	**<0.001**	**<0.001**
**Unable to get air through the nose during exercise**	3.13 (0.83)	1.36 (0.94)	0.14 (0.35)	**<0.001**	**<0.001**
**Confidence to get and keep an erection**	2.98 (1.08)	4.09 (1.02)	4.86 (0.35)	**<0.001**	**<0.001**
**Erection hard enough for penetration**	2.92 (1.53)	3.70 (1.31)	4.96 (0.20)	**<0.001**	**<0.001**
**Able to maintain the erection after penetration**	2.87 (0.88)	3.53 (0.93)	4.98 (0.14)	**<0.001**	**<0.001**
**Difficult to maintain the erection to completion of intercourse**	3.45 (0.91)	3.89 (1.01)	4.94 (0.24)	**<0.001**	**<0.001**
**Grade of satisfaction**	4.13 (0.68)	4.32 (0.64)	4.94 (0.24)	**0.015**	**<0.001**

**Table 3 jcm-12-06626-t003:** Mixed-effect model outcome. “**Confidence to get and keep erection**”. Mixed-effect multiple linear regression model, with time (pre/post) as a random effect. Significant values are marked in bold.

Fixed Effects	Estimate	*p*-Value
**Nasal Stuffiness**	−0.06009	0.569
**Nasal Obstruction**	0.30565	0.088
**Trouble Breathing**	−0.08294	0.619
**Trouble Sleeping**	0.16158	0.102
**Unable to get air**	0.04342	0.704

**Table 4 jcm-12-06626-t004:** Mixed-effect model outcome. “**Erection hard enough for penetration**”. Mixed-effect multiple linear regression model, with time (pre/post) as a random effect. Significant values are marked in bold.

Fixed Effects	Estimate	*p*-Value
**Nasal stuffiness**	−0.2335	0.080
**Nasal obstruction**	0.5082	**0.024**
**Trouble breathing**	−0.2187	0.299
**Trouble sleeping**	−0.3360	**0.008**
**Unable to get air**	0.4056	**0.006**

**Table 5 jcm-12-06626-t005:** The mixed-effect model outcome “**Able to maintain the erection after penetration**”. Mixed-effect multiple linear regression model, with time (pre/post) as a random effect. Significant values are marked in bold.

Fixed Effects	Estimate	*p*-Value
**Nasal stuffiness**	0.13764	0.135
**Nasal obstruction**	0.22629	0.144
**Trouble breathing**	−0.02678	0.853
**Trouble sleeping**	0.01877	0.826
**Unable to get air**	0.04239	0.670

**Table 6 jcm-12-06626-t006:** Mixed-effect model outcome “**Difficult to maintain erection to completion of intercourse**”. Mixed-effect multiple linear regression model, with time (pre/post) as a random effect. Significant values are marked in bold.

Fixed Effects	Estimate	*p*-Value
**Nasal stuffiness**	−0.09336	0.308
**Nasal obstruction**	0.02070	0.876
**Trouble breathing**	−0.15772	0.302
**Trouble sleeping**	−0.13655	0.131
**Unable to get air**	0.09636	0.327

**Table 7 jcm-12-06626-t007:** The mixed-effect model outcome “**Grade of satisfaction**”. Mixed-effect multiple linear regression model, with time (pre/post) as a random effect. Significant values are marked in bold.

Fixed Effects	Estimate	*p*-Value
**Nasal stuffiness**	−0.07023	0.266
**Nasal obstruction**	0.38287	**<0.001**
**Trouble breathing**	−0.09171	0.358
**Trouble sleeping**	−0.04029	0.492
**Unable to get air**	0.09166	0.181

## Data Availability

Data are available in MDPI.

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
