# Peer review of "Is There a Correlation between Endoscopic Sinus Surgery and Improvement in Erectile Dysfunction?"

_jcm, 2023, doi:10.3390/jcm12206626_

Round 1
Reviewer 1 Report
This is an interesting study with good results that add to the literature. It is well executed.
My only comment would be to ask the authors to comment on whether a similar outcome has been seen in female patients?
Author Response
Thank you for your valuable comments. The IIEF-5 score cannot be applied to female for anatomical reasons. About the NOSE score, yes, also female reported a positive outcome in the postoperative phase due to a bettering in breathing and ventilation.
Reviewer 2 Report
Thank you for inviting me to review an interesting manuscript titled „Is there a correlation between endoscopic sinus surgery and improvement of erectile dysfunction”. In this prospective study involving 53 patients, the authors investigated whether the improvement in chronic obstructive upper airway disease correlated with the improvement in erectile function in Italian men.
Line 59 and the abstract inform that this is a prospective study, but the description of methods indicates its retrospective character. Methods read that “patients were selected from hospital archives” and that the IIEF-5 and NOSE questionnaires “were submitted routinely during the first anamnesis and at 3 months follow up after surgery”. Please clarify this issue.
Can you add characteristics of the control group and present a comparison between the study group and the control group?
Can you please add informative titles to the tables? A title such as “Data described as means ± SD” is not informative. Actually, this information should go into the table or be added in the footnote. The title should inform about the content of the table. These comments relate to all visuals.
Line 127. The sentence “From Table 1 it is evident which was the impact of endoscopic sinus surgery on the patient's erectile dysfunction” is inaccurate. First, the impact can be assessed by a regression analysis, not by comparison. Second, this sentence belongs more to the discussion, as it contains the authors' opinion.
Line 178 lacks a reference. None of the references include a study by Fidan.
Line 225, what is the rationale behind discussing high BMI while patients with obesity were excluded? The same comment applies to other risk factors that served as exclusion criteria.
Lines 235-238, what knowledge does your study add regarding NO, CRP, and TNF-α that makes this statement reasonable in your discussion?
Generally, the discussion is difficult to follow. There are repetitions – for example, 188-192 and 204-206 in the next paragraph present the same messages. It would be good to group factors contributing to ED and describe how they act in separate paragraphs. In the paragraph starting with line 175, you start from nasal obstruction, go through vascular factors, go back to nasal airway obstruction and end with olfactory regulation. The next paragraph starting with line 204, again goes to vascular factors.
The discussion presents many risk factors for ED; however; hormonal disturbances, particularly testosterone, are omitted. Did you check testosterone levels in those patients who had ED? Please see how important is to check testosterone levels in patients with ED in the paper by Kalka et al. titled “Diagnosis of hypogonadism…” Transl Androl Urol 2020 Dec;9(6):2786-2796.
Line 256, ED can be divided into organic and psychogenic. In the context of ED classification, this sentence sounds inaccurate. Please rephrase. You can support the classification of ED with an article by Pozzi et al. “Primary organic versus primary psychogenic erectile dysfunction...” Andrology. 2022 Oct;10(7):1302-1309. There are many articles on the pathogenesis of organic ED. You may want to look at this one as an example Diehm et al. “Interdisciplinary options for diagnosis and treatment of organic erectile dysfunction” Swiss Med Wkly. 2015,28;145:w14268.
What are the limitations of your study?
There are problems with capitalization, eg. names of diseases should be spelt with small letters.
Abbreviations should be defined at first mention and used consistently after that. There are some inconsistencies, e.g. lines 17 and 56. In line 47, CSR is defined for the second time. Furthermore, an abbreviation should be introduced when used at least 3 times. In lines 36 and 50, abbreviations are used unnecessarily.
Please use dots, not commas for decimals.
Punctuation, typos, capitalisation and minor grammatical errors must be corrected.
Author Response
- Nature of the study has been clarified.
- Characteristics of the control group and present a comparison are now added in material in methods.
- Informative data to table have been added.
- This sentence has been modified and clarified.
- Reference has been added.
- More information about BMI and risk factors is now clarified in the paragraph Discussion.
- This statement has been modified.
- Discussion has been reorganized.
- Thank you for providing this precious suggestion. We didn’t check testosterone level in this preliminary study. Considering the promising results, we are already planning to include this analysis in the next study.
- ED classification has been completely reorganized.
- Limitations of the study have been clarified.
- 13. 14. All graphic mistakes have been corrected.
Reviewer 3 Report
1. In results of abstract part, "overall satisfaction at the conclusion of intercourse (p-value <0.001) "should be written as (p-value < 0.001)
2. In methods part, “Male patients >18 years old” was controversial with patients’ age enrolled in the research (aged between 40 and 70).
3. Please added the statistical table of IIEF5 scores for 53 patients, including the scores and baseline clinical data
4. The author's description of the results is too simple. If possible, the authors should add IIEF5 evaluation and record different follow-up times, to make the results more accurate.
5. The use of the IIEF5 Score alone is insufficient to assess erectile dysfunction, it is recommended to add the Erection Hardness Score and others.
6. Improving the patient's ventilation status might have an impact on the patient's abnormal nocturnal erection stat, I hope the author can discuss it.
7. The reference is too old, please update the reference of chronic rhinosinusitis on sleep-disordered breathing
Author Response
- P-value has been corrected.
- Age has been corrected.
- Table with IIEF-5 score has been included.
- The IIEF-5 evaluation was made in correlation with the complete recovery from FESS surgery. Thank you for the suggestion, we will consider it for further studies.
- As a preliminary study, we adopted the IIEF-5 score as an already validates scale used in different fields as endocrinology, orthopedics, etc. Ex: Kumar R, Kumar U, Trivedi S. Comparison of Risk Factors for Erectile Dysfunction (ED) in Type 2 Diabetics and Nondiabetics: A Retrospective Observational Study. Cureus. 2023 Sep 2;15(9):e44576. doi: 10.7759/cureus.44576. PMID: 37790032; PMCID: PMC10545003. --- Alkan H, ErdoÄŸan Y, Veizi E, Sezgin BS, Çepni Åž, Mert HÇ, Fırat A. Better sex after hip arthroscopy; Sexual dysfunction in patients with femoroacetabular impingement syndrome. Orthop Traumatol Surg Res. 2023 Sep 28:103693. doi: 10.1016/j.otsr.2023.103693. Epub ahead of print. PMID: 37776950.
- Sadly, we don’t have the device that could allow us to evaluate the Nocturnal penile tumescence testing. But considering the results obtained from the study, we are thinking to implement it in our diagnostic path.
- Reference have been implemented and updated.
Round 2
Reviewer 2 Report
The comments have been addressed.
The manuscript requires further proofreading to eliminate grammar errors e.g. line 64 "A total of 280 patients were selected from the patients were selected during the ".
Language harmonisation is needed, please choose either American or British. you are using both spellings, e.g. line 322 - aetiology, line - 328 etiology.
Author Response
Dear Reviewer,
Deep grammar and spell check has been done.
Thank you for your comments!